# Wind Speed Retrieval Using Global Precipitation Measurement Dual-Frequency Precipitation Radar Ka-Band Data at Low Incidence Angles

Chong Jiang [1,2], Lin Ren [2,3], Jingsong Yang [2,3,*], Qing Xu [1] and Jinyuan Dai [2]

1 College of Oceanography, Hohai University, 1 Xikang Road, Nanjing 210024, China; fozen@hhu.edu.cn (C.J.); maggiexu@hhu.edu.cn (Q.X.)
2 State Key Laboratory of Satellite Ocean Environment Dynamics, Second Institute of Oceanography, Ministry of Natural Resources, 36 Baochubei Road, Hangzhou 310012, China; renlin210@sio.org.cn (L.R.); daijinyuan@sio.org.cn (J.D.)
3 Southern Marine Science and Engineering Guangdong Laboratory (Zhuhai), 1 Jintang Road, Zhuhai 519082, China
* Correspondence: jsyang@sio.org.cn; Tel.: +86-0571-8196-3111

**Abstract:** In this study, sea surface wind speed was retrieved using the Global Precipitation Measurement (GPM) dual-frequency precipitation radar (DPR) Ka-band data. In order to establish the Ka-band model at low incidence angles, the dependence of the DPR Ka-band normalized radar cross section (NRCS) on the wind speed, incidence angle, sea surface temperature (SST), significant wave height (SWH), and sea surface current speed (CSPD) was analyzed first. We confirmed that the normalized radar cross section depends on the wind speed, incidence angle, and SST. Second, an empirical model at low incidence angles was established. This model links the Ka-band NRCS to the incidence angle, wind speed, and SST. Additionally, the wind speed was retrieved by the model and was validated via the GPM Microwave Imager (GMI) wind product. The validation yielded a root mean square error (RMSE) of 1.45 m/s and the RMSE was better at a lower incidence angle and a higher SST. This model may expand the use of GPM DPR data in enriching the sea surface wind speed data set. It is also helpful for other Ka-band spaceborne radars at low incidence angles to measure wind speed in the future.

**Keywords:** global precipitation measurement dual-frequency precipitation radar; Ka-band; low incidence angles; wind speed retrieval; SST





## 1. Introduction

Sea surface wind speed is an important dynamic environmental parameter for understanding the physical processes occurring on the sea surface. Wind products are closely related to human activities such as nearshore engineering or fisheries and can even be utilized for renewable energy generation [1]. Currently, there are multiple active microwave radars for sea surface wind speed measurements. Spaceborne radars aimed at sea surface wind measurements include scatterometers, synthetic aperture radar (SAR), and altimeters. Scatterometers and SAR are operated at medium incidence angles (from 20° to 60°), while an altimeter is the nadir angle instrument [2–4]. Besides the above radars, some studies on the low-incidence angles (from 0° to 10°) spaceborne radars have been carried out in recent years. These radars include the Tropical Rainfall Measuring Mission (TRMM) Precipitation Radar (PR), Global Precipitation Measurement (GPM) Dual-frequency Precipitation Radar (DPR), China France Ocean Satellite (CFOSAT) Surface Wave Investigation and Monitoring (SWIM), and Tiangong-2 interferometric imaging radar altimeter (InIRA) [5]. Besides their main functions, these radars show potential for application in wind speed retrieval based on the scattering principles. The wind products generated from these radars are essential for enriching the sea surface wind speed data set.

Previous studies have shown that sea surface scattering at low incidence angles is dominated by a quasi-specular scattering mechanism [6]. The NRCS near nadir is well approximated using a quasi-specular model as [7]:

$$\mathrm{NRCS}(\theta) \cong \rho / \mathrm{mss_{eff}} \cdot \sec^4 \theta \cdot \exp^{-\tan^2 \theta / \mathrm{mss_{eff}}} \qquad (1)$$

The parameter $\mathrm{mss_{eff}}$ refers to the effective total ocean surface slope variance related to sea surface roughness. When the wind speed changes, the sea surface roughness and the corresponding $\mathrm{mss_{eff}}$ change concomitantly. Thus, the $\mathrm{mss_{eff}}$ contains the wind speed nformation, which is the physics we can retrieve wind speed from NRCS at low incidence angles. Furthermore, the specular reflectivity $\rho$ can be obtained from the nadir Fresnel reflection coefficient $R(0°)$ by $\rho \equiv |R(0) * R(0)^*|$. Here, the R is a function of sea surface salinity, temperature, and frequency. At this point, it implies that the NRCS is related to the sea surface temperature (SST), besides the wind speed. This model in Equation (1) can qualitatively describe the parameters related to the NRCS, but it is difficult to use it to quantitatively retrieve wind speed. For that, scientists began to solve the wind speed retrieval problem in terms of data analysis and empirical models.

In the light of the launch of the TRMM PR, the NRCS trend with wind speed in the mechanism was confirmed. In particular, the relationships between TRMM PR Ku-band NRCS at low incidence angles and collocated TRMM Microwave Imager (TMI) wind speed were investigated [3,8]. They found that NRCS first increased and then gradually decreased with an increase of wind speed. Moreover, Tran et al. [9,10] studied the influence of sea surface wind speed and SWH on the NRCS of the Ku-band at low incidence angles. They reported that the sea surface wind speed is the main influencing factor that modifies the sea surface scattering. The potential applicability of low incidence angles has led to the development of empirical models for retrieving wind speed over the sea surface. Empirical models have been shown to be efficient for retrieving the wind field when applied to scatterometers, SARs, and altimeters. Ren et al. [11] proposed an empirical wind speed retrieval model (KULMOD) by analyzing the relationships between the NRCS and sea surface wind speed and incidence angle. The wind speeds were retrieved from the TRMM PR data and then validated with buoy measurements, yielding a root mean square error of 1.45 m/s. Bao et al. [12] established an empirical model by using the PR NRCS and QuikSCAT wind speeds. They retrieved the wind speed using the maximum likelihood estimation method and reported a standard deviation of 1.5 m/s compared with the buoy. Furthermore, Panfilova et al. [13] proposed an algorithm for calculating sea surface wind speed in a wide swath using GPM DPR Ku-band data. The wind speed was retrieved from the equivalent NRCS values at nadir. When comparing the retrievals with the buoy winds, the BIAS was 0.26 m/s and the standard deviation was 1.88 m/s. In these studies, it was also found that the correlation between NRCS and the wind direction was weak at low incidence angles. The wind direction was thus neglected in the wind speed retrieval model.

In previous studies on TRMM PR data [11–16], the data characteristics of Ku-band NRCS have been analyzed. Some low incidence Ku-band empirical wind speed retrieval models have been developed. In 2014, the GPM as a follow-up satellite for TRMM was launched. GPM DPR is also operated at low incidence angles but has two bands (Ku and Ka). If the radars all have accurate radiometric calibration, the wind speed retrieval model based on TRMM PR can also be applied to other satellites. For example, GPM DPR, the Chinese Tiangong-2 space laboratory (TG2) interferometric imaging radar altimeter (InIRA), and CFOSAT SWIM were considered. In this case, the Ku-band models at low incidence angles developed by TRMM PR and GPM DPR data have been used for wind speed retrieval for TG2 InIRA and CFOSAT SWIM, respectively [17]. Both approaches resulted in good retrieval accuracy (RMSE of ~1.5 m/s). With further research, scientists began to expand their research from Ku-band to Ka-band. Based on the GPM DPR data, the analysis of Ka-band has yielded many similar results to the former Ku-band in terms of the wind sensitivity of NRCS. For example, both the NRCS of two bands have monotonic decreasing trend with wind speed. However, there are also some differences between them.

Vandemark et al. [18] found that Ka-band NRCS is related to SST and that the NRCS change rate reached 15% at the lowest and highest temperature. In addition, the NRCS of Ka-band is more sensitive to sea surface state than the NRCS of Ku-band. Hossan et al. [19] derived the geophysical model functions (GMFs) for the Ku- and Ka-band over an incidence angle range of $\pm 18°$ and the SST effects on the Ku- and Ka-band NRCS were assessed. The results showed that the SST impact is greater at the Ka-band than at the Ku-band.

As noted above, some Ku-band wind speed retrieval models at low incidence angles have been established, but the Ku-band model is not suitable for Ka-band data. Furthermore, scientists have proposed the Ka-band GMF model, but it is difficult to directly use this model to retrieve wind speed in case of the absence of auxiliary collocated wind direction data [19]. In this study, the NRCS characteristics on SST, SWH, and CSPD have been studied further. On this basis, the Ka-band wind speed retrieval model is established, which uses the SST as a model parameter in addition to common wind speed and incidence angles. The wind speed retrieved by the model was validated using the GMI wind product to estimate the retrieval accuracy. The datasets and model are presented in Section 2. The results and analyses are given in Section 3. The discussion and conclusions are provided in Sections 4 and 5, respectively.

## 2. Data

The data used in this study include the GPM DPR Ka-band NRCS and collocated wind speed, SST, SWH, and CSPD. The GPM satellite was launched in February 2014 from Japan. It carried the first space-borne Ku/Ka-band dual-frequency precipitation radar [20,21]. The Ka-band precipitation radar (KaPR) is operated at 35.5 GHz, while the Ku-band precipitation radar (KuPR) is operated at 13.6 GHz. The spatial resolution of the GPM DPR is approximately 5 km (at nadir). The Ka-band has 25 scanning beams. The Ku-band has 49 scanning beams, 25 of which are matched with the Ka-band. The DPR data used in this study were attenuation-corrected Level 2A. The time range of the data was from January 2018 to June 2018. In the data preprocessing, the precipitation flag was applied to select the NRCS without rain, while the land surface type was used to classify the land surface (sea or land). The NRCS was smoothed with an average filter (about 25 km $\times$ 25 km) to effectively reduce fluctuations.

The collocated wind speed and SST were from GPM GMI with a spatial resolution of 25 km, which has a comparable resolution to the scatterometer. The collocated SWH data were from the European Center for Medium-Range Weather Forecasts (ECMWF) reanalysis datasets with a temporal resolution of 6 h and a spatial resolution of 0.125°. The collocated CSPD data were taken from the HYbrid Coordinate Ocean Model (HYCOM) global sea surface current speed. The temporal resolution was 3 h and the spatial resolution was 1/12°. Here, the wind speed and SST were satellite-derived data, while SWH and CSPD were derived from the model. For the former, the observation time is the same as that of the GPM-derived data. In the process of matching, we only needed to make the center distance between the DPR and GMI data less than 6 km. For the latter, temporal and spatial interpolation can be used to match GPM data.

The incidence angle of the DPR Ka-band ranges from $-9°$ to $9°$. In this study, the absolute value of the incidence angle was used for convenience. The influences of wind direction and current direction were not considered here. Previous studies demonstrated that the NRCS showed no clear monotonic decreasing trend at very low or high wind speeds [15]; thus, the wind speed range used in this study was 2–18 m/s. Moreover, the SST of most of the sea area was less than 30 °C from the data distribution. When the SST is below 0 °C, possible ice conditions may occur, affecting the radar NRCS; therefore, the SST range used was from 1 to 30 °C. Figure 1 shows the data distribution along the incidence angle, wind speed, and SST bin. As seen from Figure 1, the data counts are uniformly distributed with the incidence angles, while it has a maximum from 7 to 8 m/s and a minimum from 11 to 14 °C. There are $1.5 \times 10^7$ data points in total, half of which were used to establish the model, while the other half were used to validate the model. Figures 2 and 3

show the global distribution of the GMI wind speed and SST after collocated with DPR NRCS. We selected the data from 1 January to 5 January 2018 for drawing. In Figure 2, the minimum wind speed is 2 m/s and the corresponding color is dark blue; the maximum wind speed is 18 m/s and the corresponding color is yellow. The wind speed is low in low latitudes. As the latitude increases, the wind speed increases. In Figure 3, the yellow represents the sea surface temperature of 30 °C and the dark blue represents the sea surface temperature of 1 °C. The distribution of global SST is very regular. The SST is highest in the equatorial region. With the increase of latitude, the SST gradually decreases. At the poles, the sea surface temperature is the lowest.

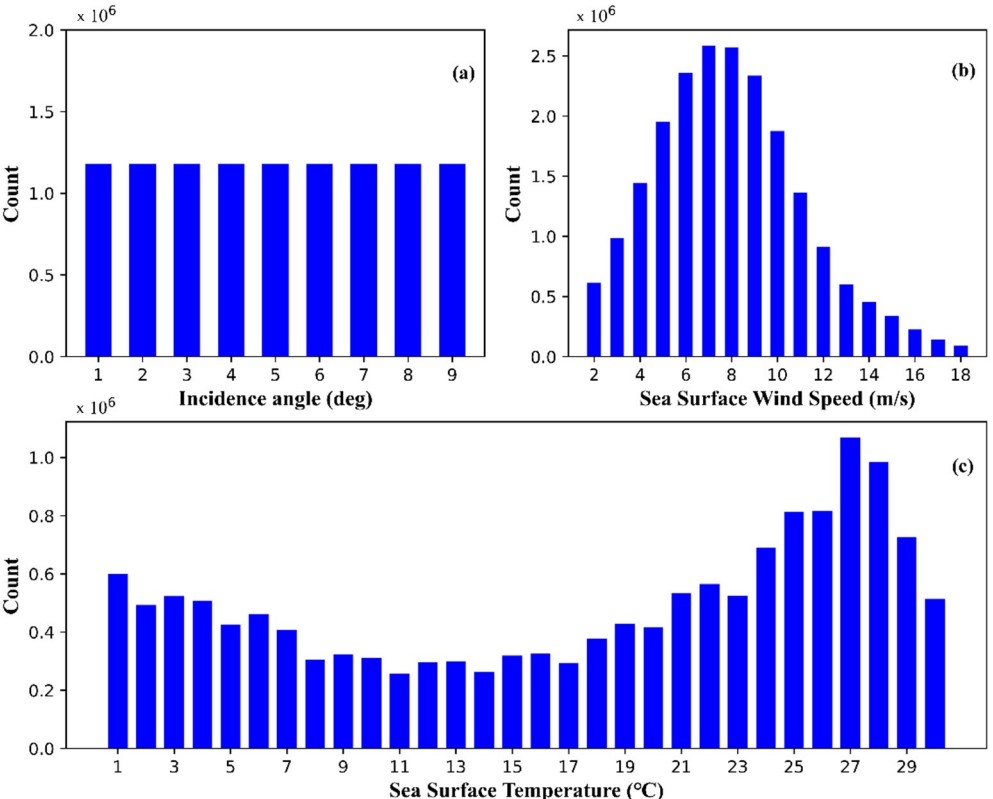

**Figure 1.** The data distribution along with (**a**) incidence angles, (**b**) wind speeds, and (**c**) SST.

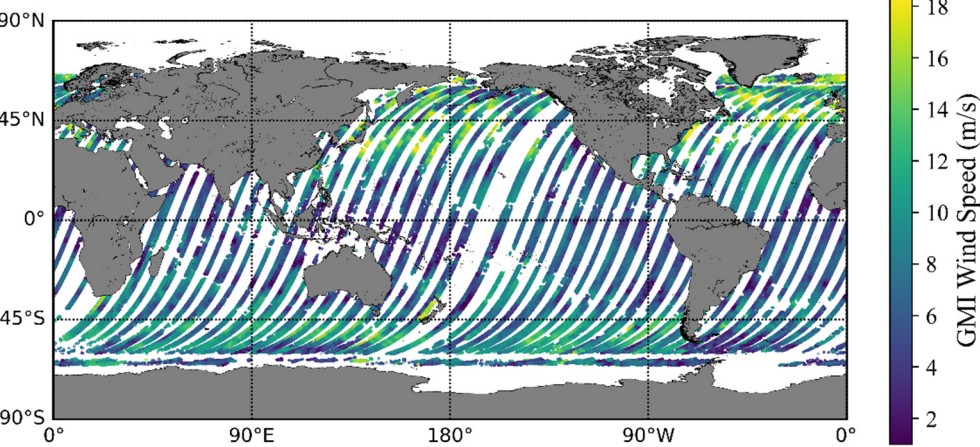

**Figure 2.** Global distribution of the GMI wind speed after collocated with DPR NRCS (From 1 January to 5 January 2018).

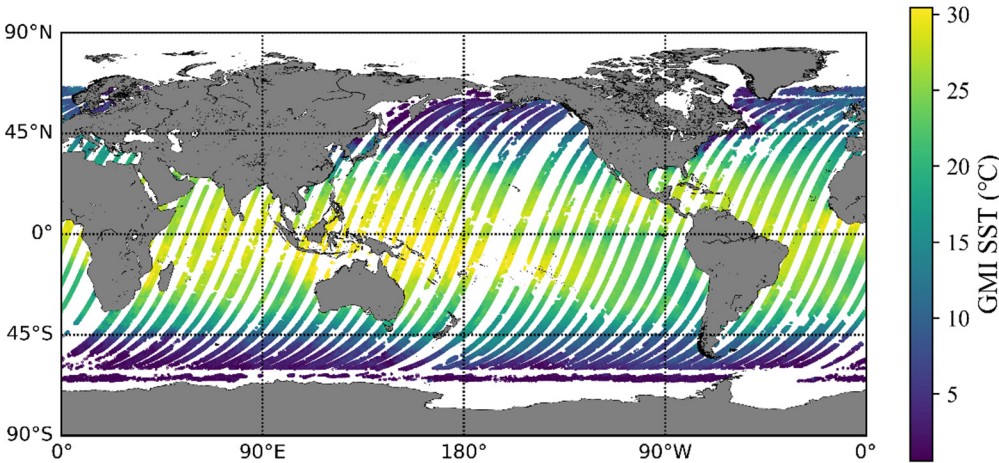

**Figure 3.** Global distribution of the GMI SST after collocated with DPR NRCS (From 1 January to 5 January 2018).

## 3. Results

### 3.1. Data Analysis

First, the relationships between the Ka-band NRCS and wind speed at each incidence angle are described in Figure 4. This figure shows that the NRCS decreases with the wind speed at each incidence angle bin. This finding is consistent with the findings of previous studies [14]. When the wind speed was fixed, the NRCS also decreased with the incidence angle. This finding indicates that the Ka-band data are similar to the Ku-band data in terms of the relationships with wind speed and incidence angle [11]. Moreover, the difference in NRCS between 2 and 18 m/s was larger when the incidence angle was smaller. For instance, at the incidence angle of 1°, the difference in NRCS between 2 and 18 m/s is about 6 dB. This corresponds to 2 dB when the incidence angle is 9°. This finding suggests that the Ka-band data at lower incidence angles are more useful for wind speed retrieval.

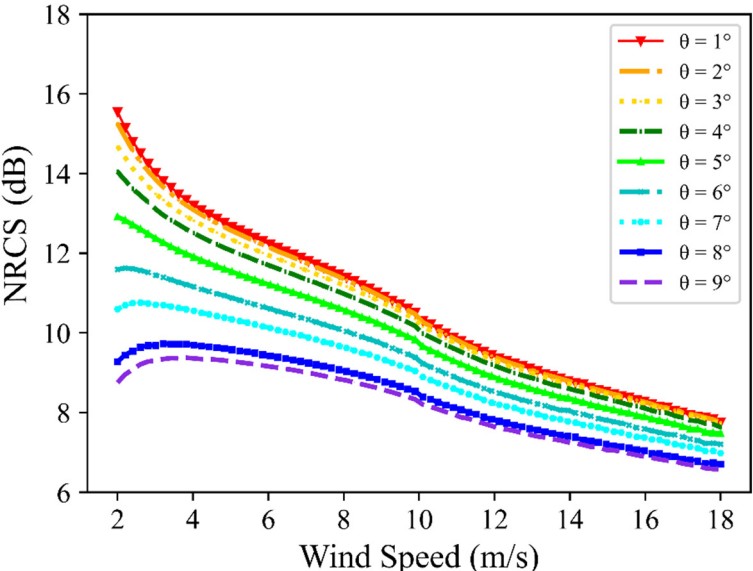

**Figure 4.** Mean value of GPM DPR Ka-band NRCS dependence on wind speed at each incidence angle bin. The symbols represent 9 different incidence angles bins ranging from 1° to 9°.

Next, in the quasi-specular model with low incidence angles, NRCS is affected by the coefficient $mss_{eff}$, which is related to sea surface roughness. As both SWH and CSPD can cause changes of sea surface roughness, this study thus researched the NRCS dependence on SWH and CSPD. In addition, scientists found that the effect of SST on Ka-band NRCS

cannot be ignored, because the SST is related to the specular reflectivity $\rho$. Therefore, this study also investigated the NRCS dependence on SST under different incidence angles and wind speeds in detail. As explained in the introduction, the influence of wind direction on NRCS is weak and not considered in this study.

The relationship between NRCS and SWH is analyzed in Figure 5, in which the incidence angle and wind speed are fixed. In the figure, the red area indicates the points that are denser, while the green area represents the points that are sparser. As seen in this figure, when the wind speed is 4 m/s and 8 m/s the NRCS will gradually decrease with the increase of SWH; at this time, the correlation coefficient between SWH and NRCS is about −0.3. As the wind speed increases to 12 m/s, the correlation between SWH and NRCS gradually decreases. The data distribution also gradually becomes "round". This also shows that the sensitivity of NRCS to SWH decreases with the increase of wind speed. Furthermore, if SWH is considered in the model, collocated SWH data sources are needed for the retrieval process, which would lead to limitations. Therefore, the influence of SWH on NRCS is not considered in the wind speed retrieval model in this study due to the weak correlation and convenience of retrieval.

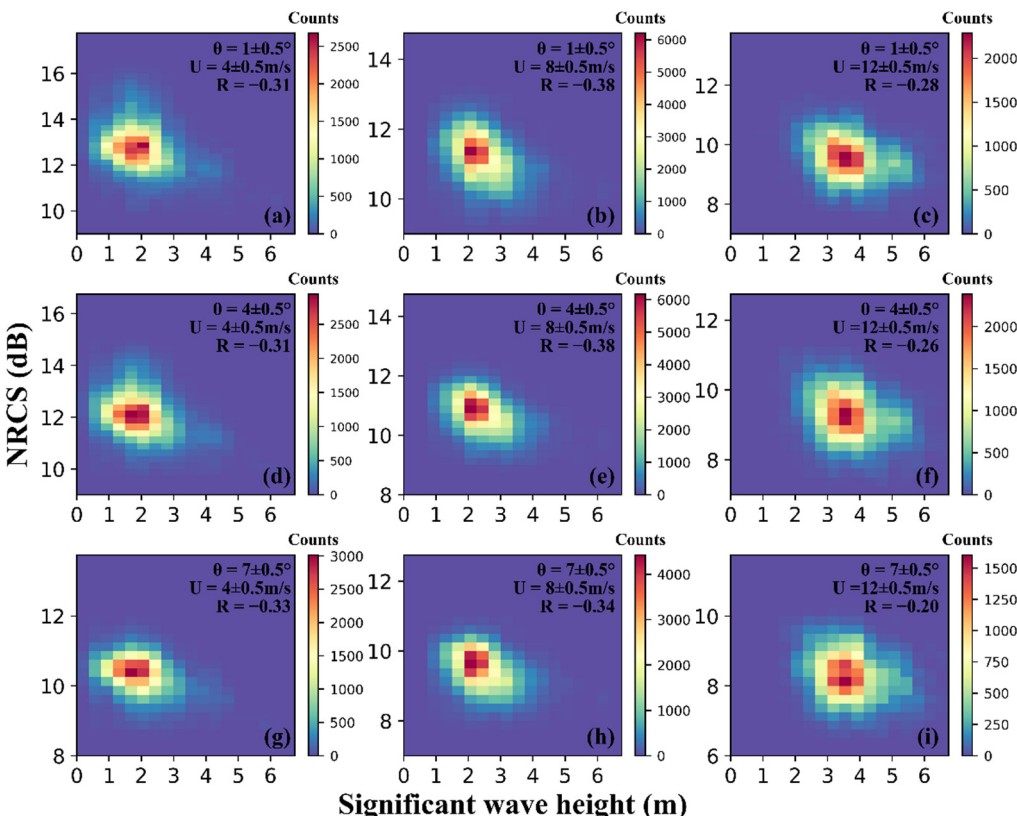

**Figure 5.** GPM Ka-band NRCS dependence on SWH at selected incidence angle and wind speed: (**a**) $1 \pm 0.5°$ and $4 \pm 0.5$ m/s; (**b**) $1 \pm 0.5°$ and $8 \pm 0.5$ m/s; (**c**) $1 \pm 0.5°$ and $12 \pm 0.5$ m/s; (**d**) $4 \pm 0.5°$ and $4 \pm 0.5$ m/s; (**e**) $4 \pm 0.5°$ and $8 \pm 0.5$ m/s; (**f**) $4 \pm 0.5°$ and $12 \pm 0.5$ m/s; (**g**) $7 \pm 0.5°$ and $4 \pm 0.5$ m/s; (**h**) $7 \pm 0.5°$ and $8 \pm 0.5$ m/s; (**i**) $7 \pm 0.5°$ and $12 \pm 0.5$ m/s. R represents the correlation coefficient between SWH and NRCS. The NRCS range used to plot is 6–18 dB.

The analysis of the relationship between NRCS and CSPD is described in Figure 6, which adopts a similar approach shown in Figure 5. In this study, we did not consider the wind direction; here, the relative direction angle between wind direction and flow direction is not further considered. As shown, the NRCS does not exhibit any linear change with the CSPD. The correlation coefficient between SWH and NRCS is about 0.1. This indicates that CSPD has little effect on the NRCS and the CSPD is thus not used as the model input.

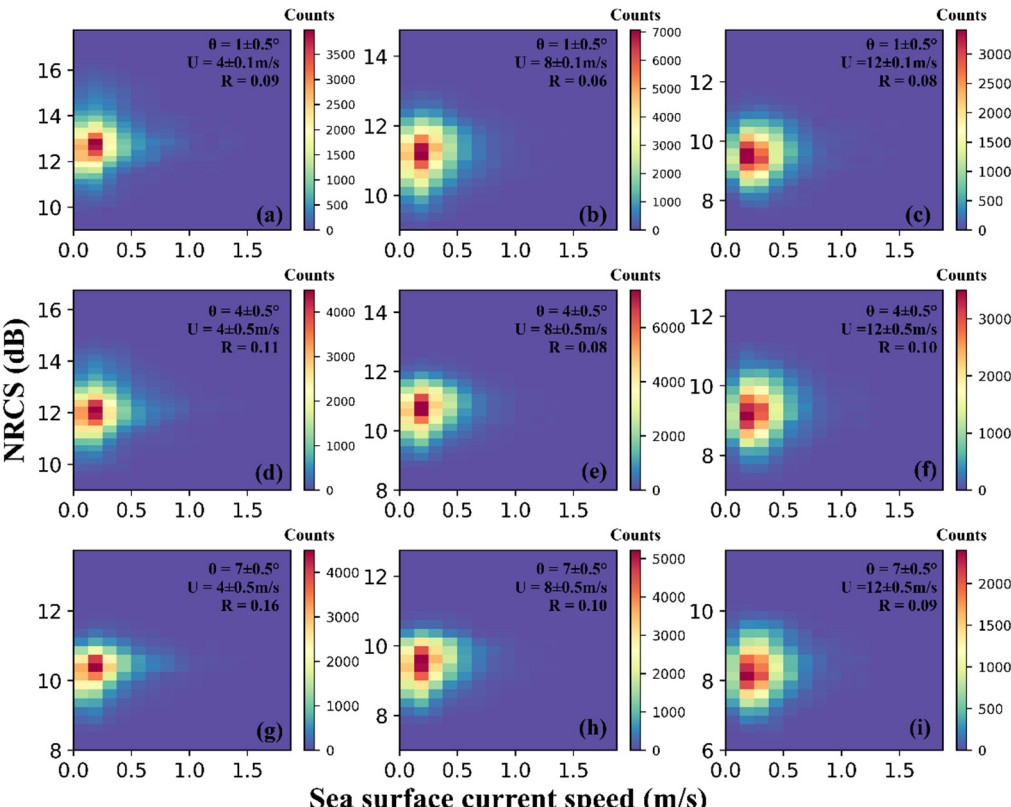

**Figure 6.** GPM Ka-band NRCS dependence on CSPD at selected incidence angle and wind speed: (**a**) 1 ± 0.5° and 4 ± 0.5 m/s; (**b**) 1 ± 0.5° and 8 ± 0.5 m/s; (**c**) 1 ± 0.5° and 12 ± 0.5 m/s; (**d**) 4 ± 0.5° and 4 ± 0.5 m/s; (**e**) 4 ± 0.5° and 8 ± 0.5 m/s; (**f**) 4 ± 0.5° and 12 ± 0.5 m/s; (**g**) 7 ± 0.5° and 4 ± 0.5 m/s; (**h**) 7 ± 0.5° and 8 ± 0.5 m/s; (**i**) 7 ± 0.5° and 12 ± 0.5 m/s. R represents the correlation coefficient between CSPD and NRCS. The NRCS range used to plot is also 6–18 dB.

Figure 7 shows the analysis of the influence of SST on the NRCS. The same incidence angle and wind speed were used. It shows that the NRCS increases with the SST at different incidence angles and wind speeds. For example, when the SST are 1 °C and 30 °C, at the conditions of 4° and 7 m/s, the NRCS difference between them is about 1.0 dB. In Figure 7a, when the incidence angle is 1° and the wind speed is 4 m/s, the data are more discrete and there are many large NRCS values deviated from the main trend. This is mainly because when the wind speed and incidence angle are all low, the sea surface is relatively smooth and the radar beam is approximately perpendicular to the sea surface. Due to strong specular reflection, some NRCS values (the radar beam is perpendicular to small facet) will be abnormally large. When the incidence angle and the wind speed become larger, this phenomenon gradually disappears and the NRCS value consequently becomes more concentrated. The correlation coefficient between SST and NRCS is from 0.25 to 0.54. Compared with SWH and CSPD, the correlation between SST and Ka-band NRCS is relatively higher. This confirms that the SST is significantly related to Ka-band NRCS and should be introduced into the wind speed retrieval model.

### 3.2. Model Design

In the previous Ku-band low incidence wind speed retrieval model, only the NRCS, wind speed, and incidence angle were utilized as the input [11]. The analysis of the previous section indicated that the Ka-band wind speed retrieval model should use wind speed, incidence angle, and SST as the model input. To quantify the impact of the SST factor in the Ka-band model, we analyzed the influence of SST on the relationship between the NRCS and wind speed. Figure 8 shows the NRCS variations with wind speed at five SST bins when the incidence angle is 4°. As shown in the figure, the NRCS under a higher

temperature is greater than that under a low temperature in the wind speed range of 2 to 18 m/s. Furthermore, the NRCS trends at different SST are nearly parallel.

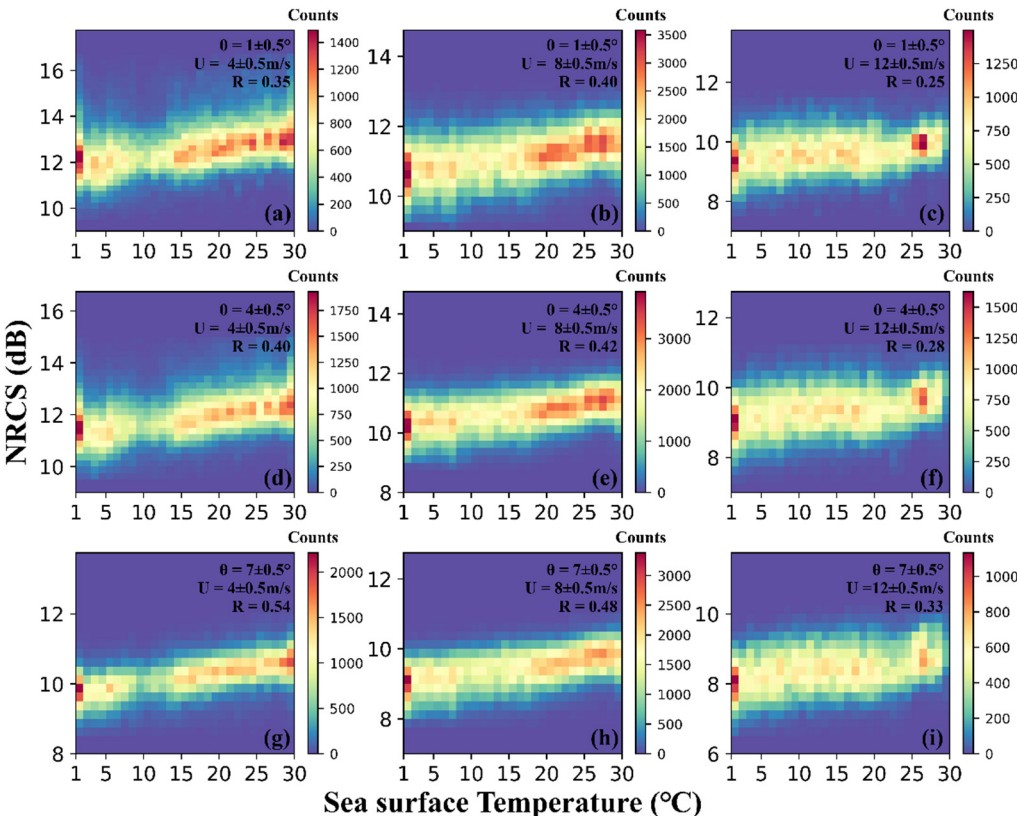

**Figure 7.** GPM Ka-band NRCS dependence on SST at selected incidence angle and wind speed: (**a**) $1 \pm 0.5°$ and $4 \pm 0.5$ m/s; (**b**) $1 \pm 0.5°$ and $8 \pm 0.5$ m/s; (**c**) $1 \pm 0.5°$ and $12 \pm 0.5$ m/s; (**d**) $4 \pm 0.5°$ and $4 \pm 0.5$ m/s; (**e**) $4 \pm 0.5°$ and $8 \pm 0.5$ m/s; (**f**) $4 \pm 0.5°$ and $12 \pm 0.5$ m/s; (**g**) $7 \pm 0.5°$ and $4 \pm 0.5$ m/s; (**h**) $7 \pm 0.5°$ and $8 \pm 0.5$ m/s; (**i**) $7 \pm 0.5°$ and $12 \pm 0.5$ m/s. R represents the correlation coefficient between SST and NRCS. The NRCS range used to plot is also 6–18 dB.

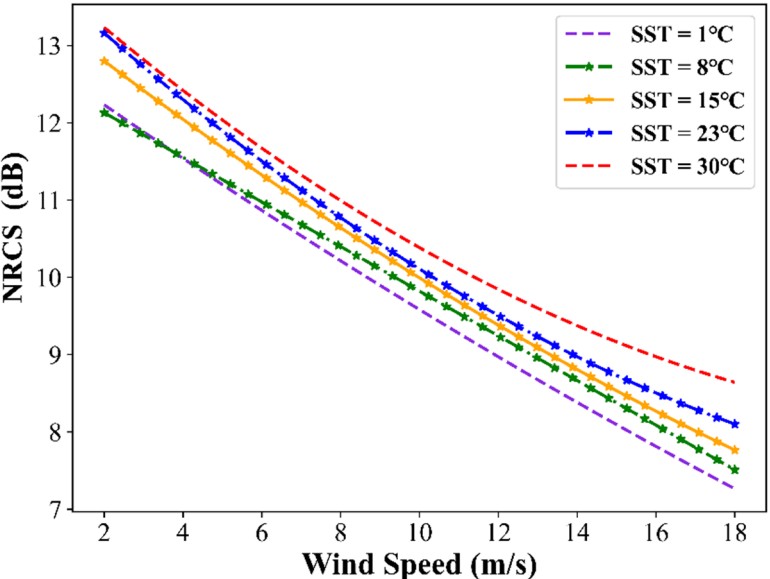

**Figure 8.** GPM Ka-band NRCS dependence on wind speed at 4° incidence angle. The symbols show 1 °C, 8 °C, 15 °C, 23 °C, and 30 °C SST bins.

Inspired by the approximate parallelism trend, the Ka-band model adopts the form of an SST-segmented function. For the backbone of the model, the model form mainly refers to the method of the Ku-band model in Ren et al. [11,22]. Thus, the Ka-band low incidence scatter model was developed as an SST-segmented linear second-order polynomial function of the incidence angle and wind speed as shown below:

$$\sigma_0\left(\theta, U, SST_{seg}\right) = a\left(\theta, SST_{seg}\right) + b\left(\theta, SST_{seg}\right)U + c\left(\theta, SST_{seg}\right)U^2 \tag{2}$$

With

$$a\left(\theta, SST_{seg}\right) = a_0 + a_1\theta + a_2\theta^2 \tag{3}$$

$$b\left(\theta, SST_{seg}\right) = b_0 + b_1\theta + b_2\theta^2 \tag{4}$$

$$c\left(\theta, SST_{seg}\right) = c_0 + c_1\theta + c_2\theta^2 \tag{5}$$

where $a_0$, $a_1$, $a_2$, $b_0$, $b_1$, $b_2$, $c_0$, $c_1$, and $c_2$ are the model coefficients and $SST_{seg}$ is the center position of the segment.

Figure 8 shows that although the NRCS increases with the SST, the NRCS increment from 1 to 30 °C at most wind speeds is less than 1 dB. The number of segments was thus set to five, and the $SST_{seg}$ was 1 °C, 8 °C, 15 °C, 23 °C, and 30 °C, respectively. In this case, the model coefficients for any SST can be derived through the interpolation method. Given this rationale, the model coefficients at five $SST_{seg}$ points were fitted using the method described by Ren et al. [11]. Furthermore, Figure 9 shows the data scatter between the DPR NRCS and the collocated GMI wind speed. The corresponding incidence angle and SST were 1° and 1 °C, respectively. By fitting the model coefficients in Figure 9, $a\left(1°, 1 °C\right)$, $b\left(1°, 1 °C\right)$, and $c\left(1°, 1 °C\right)$ can be estimated. Following a similar procedure, $a$, $b$, and $c$ at each incidence angle bin and 1 °C are derived. In this case, $a_0$, $a_1$, $a_2$, $b_0$, $b_1$, $b_2$, $c_0$, $c_1$, and $c_2$ at 1 °C can be derived by fitting $a$, $b$, and $c$ to the incidence angles. The final model coefficients at 1 °C and the other $SST_{seg}$ are listed in Table 1. Moreover, to prove the advantages of an SST-dependent model, an SST-independent model was also established. The model coefficients are listed in Table 2.

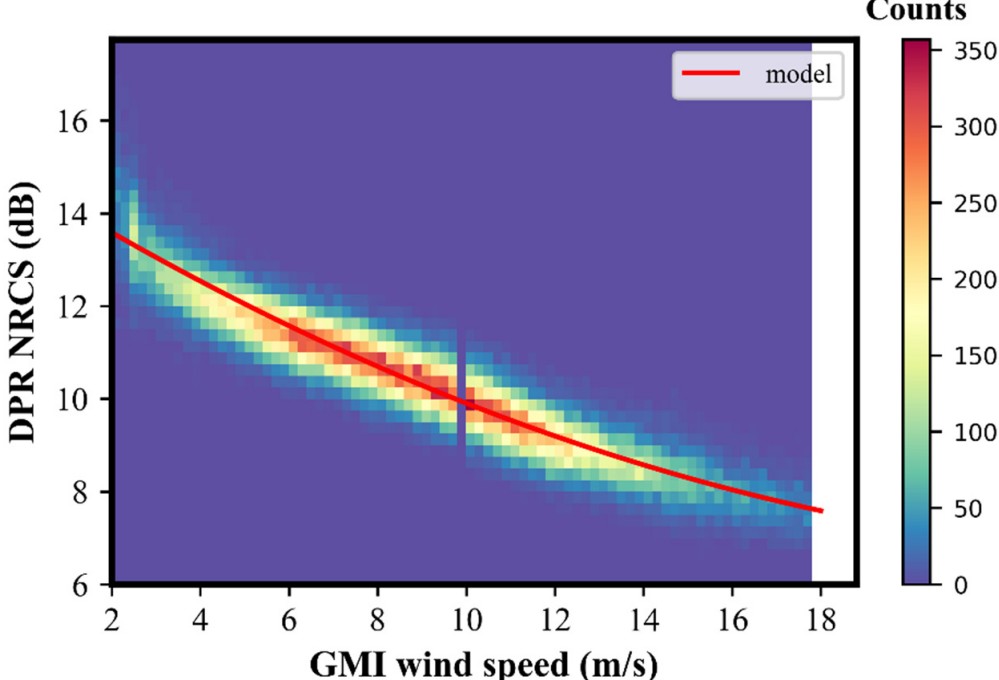

**Figure 9.** Probability density distribution of GMI wind speed and DPR NRCS at 1° incidence angle and 1 °C. The red line is the fitting for the data. The wind speed bin is 0.2 m/s and the NRCS bin is 0.25 dB.

**Table 1.** Model coefficients of the Ka-band SST-dependent wind speed retrieval model.

| SST | $a_0$ | $a_1$ | $a_2$ | $b_0$ | $b_1$ | $b_2$ | $c_0$ | $c_1$ | $c_2$ |
|---|---|---|---|---|---|---|---|---|---|
| 1 °C | 15.2450 | −0.2689 | −0.0502 | −0.6468 | 0.0351 | 0.0034 | 0.0125 | −0.0012 | −0.00009 |
| 8 °C | 15.8462 | −0.3166 | −0.0488 | −0.7088 | 0.0434 | 0.0032 | 0.0149 | −0.0015 | −0.00009 |
| 15 °C | 16.2395 | −0.3393 | −0.0495 | −0.7403 | 0.0457 | 0.0034 | 0.0160 | −0.0015 | −0.00010 |
| 23 °C | 17.1693 | −0.4589 | −0.0451 | −0.8603 | 0.0683 | 0.0030 | 0.0210 | −0.0025 | −0.00004 |
| 30 °C | 17.1002 | −0.3880 | −0.0498 | −0.8456 | 0.0566 | 0.0032 | 0.0206 | −0.0022 | −0.00004 |

**Table 2.** Model coefficients of the Ka-band SST-independent wind speed retrieval model.

| Coefficient | $a_0$ | $a_1$ | $a_2$ | $b_0$ | $b_1$ | $b_2$ | $c_0$ | $c_1$ | $c_2$ |
|---|---|---|---|---|---|---|---|---|---|
| value | 18.5516 | −0.7857 | −0.0452 | −1.1900 | 0.1429 | 0.0023 | 0.0353 | −0.0061 | −0.00004 |

*3.3. Model Validation*

Based on the developed model, this study used the lookup table method to retrieve the wind speed. The lookup table is established as follows: First, an empty table is created with the incidence angle as the row and the wind speed as the column. Using the model coefficients under 1 °C in Table 1 and the incidence angle and wind speed corresponding to each point in the lookup tables, bring into Formulas (2)–(5) to figure out the simulated NRCS. Through the same steps, the lookup tables of 8 °C, 15 °C, 23 °C, and 30 °C can be completed, respectively. Subsequently, the NRCS values in the 1 °C and 8 °C lookup tables are linearly interpolated to calculate the corresponding NRCS in the 2–7 °C lookup tables. According to the same method, the corresponding lookup tables under 1–30 °C can also be completed. After the lookup table is built, the corresponding lookup table can be selected according to the SST of the data. The wind speed solution can then be found according to the incidence angle and NRCS.

Based on the established lookup table, the wind speeds are retrieved from GPM DPR data. The retrieved wind speeds are compared with GMI wind product. Figures 10 and 11 illustrate the global distribution of the wind speed retrieved from the SST-independent model and the SST-dependent model. The same label is used in the two figures, and the wind speed range is 2–18 m/s. Blue represents the lowest wind speed and yellow represents the highest wind speed. The same part of data as Figure 2 was used. On the whole, the trend of global wind speed retrieved by the SST-independent model and the SST-dependent model are basically consistent with Figure 2; both of them have the ability to retrieve the sea surface wind speed at low incidence angles in Ka-band. Next, we will further compare the wind speed inversion accuracy of the two models.

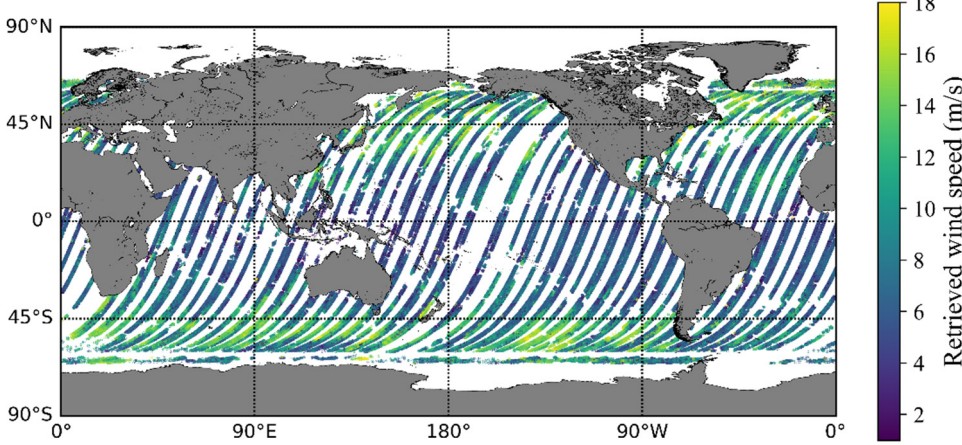

**Figure 10.** The global distribution of the sea surface wind speed retrieved by the SST-independent model (From 1 January to 5 January 2018).

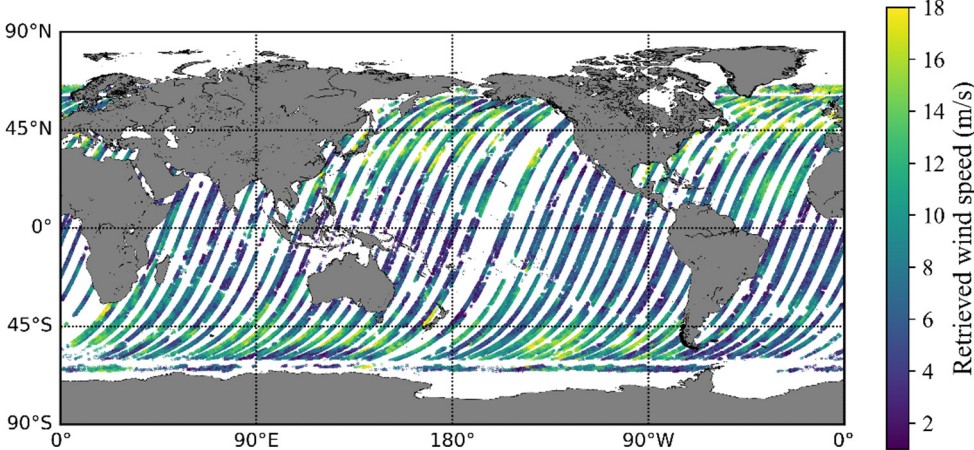

**Figure 11.** The global distribution of the sea surface wind speed retrieved by the SST-dependent model (From 1 January to 5 January 2018).

Figure 12a shows that the BIAS and the RMSE of the SST-independent model were −0.15 m/s and 1.57 m/s, while Figure 12b shows the BIAS and the RMSE of the SST-dependent model were −0.07 m/s and 1.45 m/s. As seen from these figures, both models can achieve good wind speed retrieval accuracies. Furthermore, the SST-dependent model has notably improved the BIAS and the RMSE compared to the SST-independent model. This finding confirms that the SST factor can improve the accuracy of wind speed retrieval using Ka-band data. In Figure 12a, when the retrieved wind speed is 6.3 m/s, the corresponding GMI wind speed ranges from 2.5 m/s to 10 m/s. Similarly, when the retrieved wind speed is 4 m/s, the corresponding GMI wind speed also has a large range from 2 m/s to 9 m/s. After analysis, we find that this is mainly because the sensitivity of NRCS to wind speed is significantly reduced when the incidence angles are 8° and 9°. Therefore, the Ka-band SST-independent model might retrieve many different data points into the same wind speed. When the SST-dependent model is used, due to the need to further determine the SST of data points, the fluctuation of data points will be reduced, and the fitting model under corresponding temperature will be more accurate. Therefore, no similar phenomenon occurs in Figure 12b. However, the incidence angles of 8° and 9° are still not a reliable choice for retrieving wind speed.

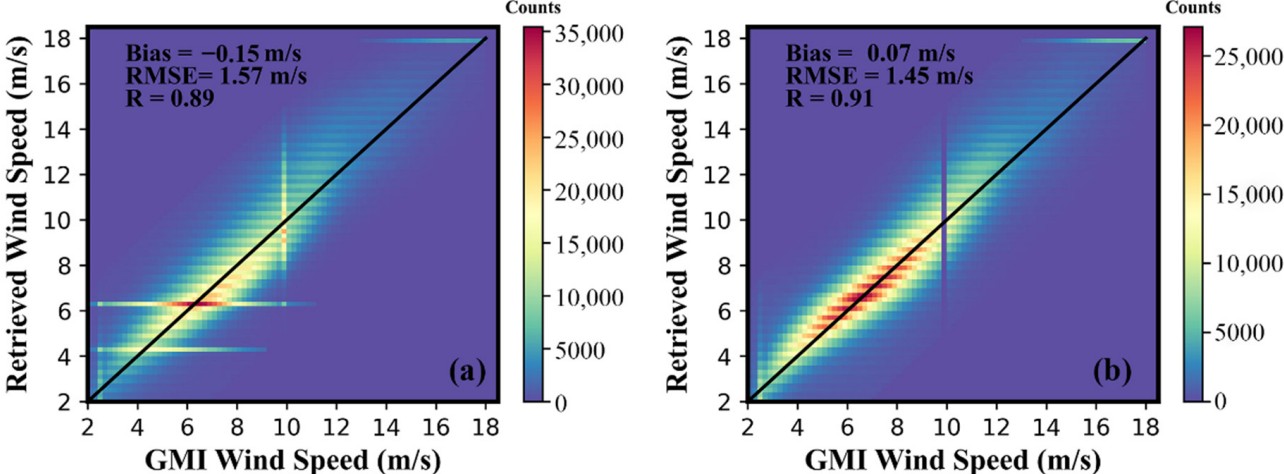

**Figure 12.** The comparison between GMI wind speed and wind speed retrieved from DPR using Ka-band low incidence: (**a**) SST-independent model, and (**b**) SST-dependent model.

In order to obtain the multi-dimensional look at the model skill, we analyzed the wind speed retrieval accuracy under various incidence angles and SST. The BIAS and the RMSE

of two models in different incidence angles are shown in Figure 13. The red line represents the SST-independent model and the blue line represents the SST-dependent model. As shown in Figure 13a, the BIAS for the two models first increased and then decreased as the incidence angle increased. The BIAS for the SST-dependent model is in the range of −0.32 to 0.11 m/s. The BIAS from the SST-independent model exhibits more pronounced fluctuations, in which the maximum BIAS reaches −1.3 m/s at 9°. Figure 13b shows that the RMSE for the two models was basically unchanged at the incidence angles lower than 4°, then increases with the incidence angle. This is mainly because the NRCS sensitivity to wind speed gradually decreases with the incidence angle, as shown in Figure 4. For the SST-dependent model, the minimum RMSE was 0.85 m/s and the maximum RMSE was 2.02 m/s. For the SST-independent model, the minimum RMSE was 0.98 m/s and the maximum RMSE was 2.53 m/s. Moreover, the RMSE of the SST-dependent model is lower than that of the SST-independent model at each incidence angle. This suggests that the accuracy of the SST-dependent model significantly improved compared with that of the SST-independent model.

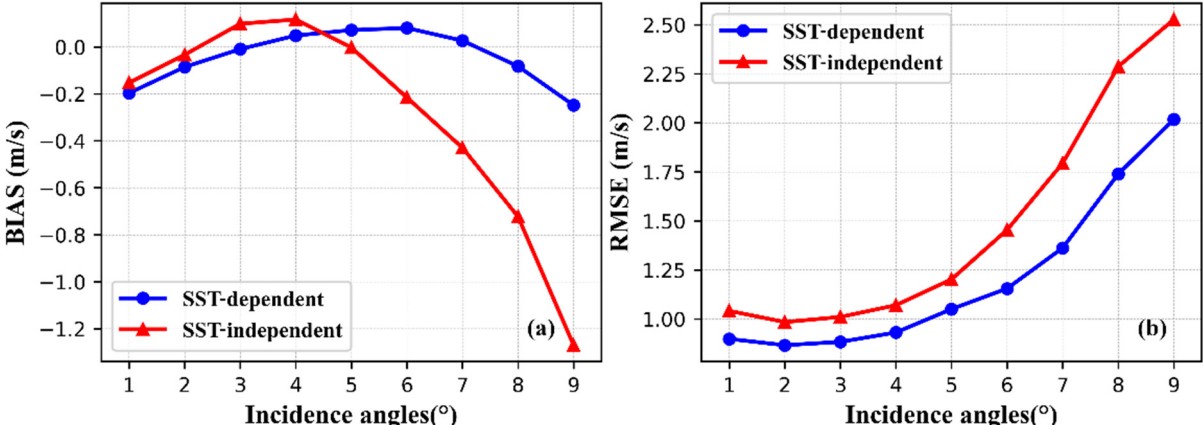

**Figure 13.** The BIAS (**a**) and RMSE (**b**) of wind speed retrieval at different incidence angles. The blue line represents the SST-dependent model, and the red line represents the SST-independent model.

The retrieval accuracy of the two models under different SST is also displayed in Figure 14. As shown in Figure 14a, the BIAS of the SST-dependent model exhibited no prominent change with the increase in SST. The BIAS is approximately −0.1 m/s. However, the BIAS of the SST-independent model significantly increased with the SST. When the SST was 1 °C, the BIAS was −1.52 m/s. When the SST was 27 °C, the BIAS was 0.51 m/s. This phenomenon is probably due to the fact that the lower SST corresponds to smaller NRCS, resulting in a higher retrieved wind speed. In this case, the retrieved wind speed is higher than the real wind speed, and the BIAS becomes negative. For a higher SST, we identified an opposite effect on the NRCS, thereby making the BIAS positive. Figure 14b demonstrates that the RMSE of both the SST-dependent model and the SST-independent model gradually decreased with an increase in SST. For the reason of this monotonic decrease, we analyzed the variance of NRCS with the different SSTs. We found that the corresponding NRCS variance gradually decreased with the SST, which is basically consistent with the trend of RMSE; therefore, we think the monotonic decrease of RMSE with SST is due to the decrease of NRCS variance. For the SST-dependent model, the maximum value of RMSE was 1.62 m/s at 1 °C, while the minimum value of RMSE was 1.21 m/s at 28 °C. For the SST-independent model, the maximum and minimum value of RMSE was 2.39 m/s at 1 °C and 1.29 m/s at 28 °C, respectively. When the SST was lower than 14 °C, the RMSE of the SST-dependent model exhibited a significant improvement compared with the SST-independent model.

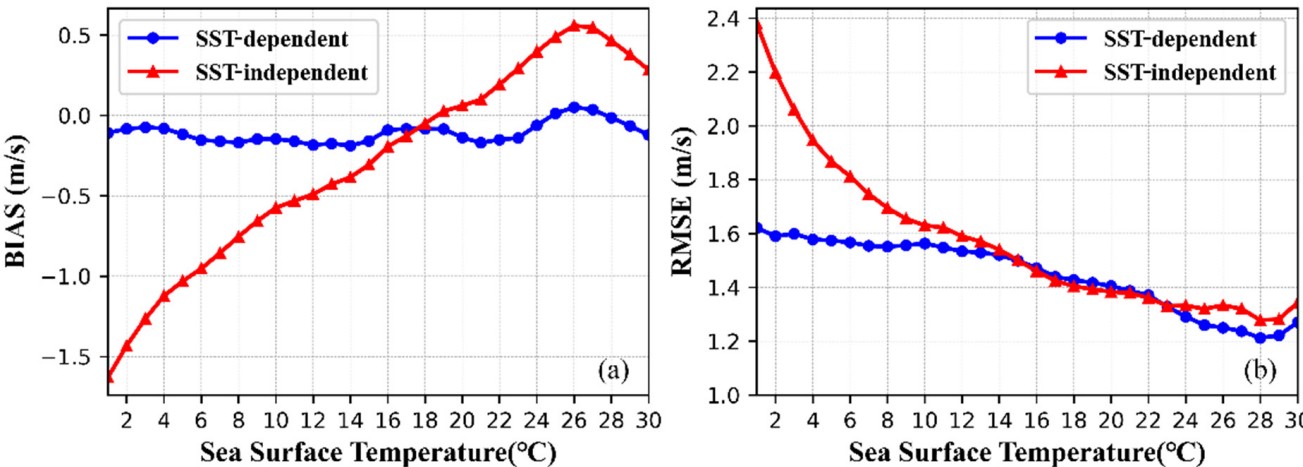

**Figure 14.** The BIAS (**a**) and RMSE (**b**) of wind speed retrievals at different SST. The blue line represents the SST-dependent model, while the red line represents the SST-independent model.

## 4. Discussion

By following a similar approach to TRMM PR, some scholars analyzed the data characteristics of GPM DPR Ku and Ka-band NRCS [18,23,24]. These analyses have demonstrated that the Ka-band NRCS is more sensitive to sea surface parameters than the Ku-band NRCS. However, for the Ka-band the NRCS characteristics related to SST and CSPD have not yet been analyzed in detail. Furthermore, a Ka-band empirical model suitable for DPR, has not yet been proposed. To alleviate this gap, our study analyzed the DPR Ka-band NRCS characteristics along with SST, SWH, and CSPD and proposed an SST-dependent low incidence Ka-band wind speed retrieval model.

The relationships between the Ka-band NRCS and SST at different incidence angles and wind speeds were analyzed. We found that the NRCS generally increased with the SST. Notably, the incidence angle and the wind speed have different effects on the relationship; in particular, this relationship was not sensitive to wind speed, yet was sensitive to the incidence angle. We showed that the lower the incidence angle the greater the influence of SST on the Ka-band NRCS. From previous studies, Ku-band NRCS with lower frequency was almost unaffected by SST changes [24]; therefore, it was expected that the NRCS at higher frequencies would be more sensitive to SST.

In addition, the influence of the SST parameter on the model accuracy was proposed; for this, a Ka-band SST-independent model was established to evaluate the SST-dependent model, whereas the BIAS and the RMSE of the two models were compared. The results showed that the BIAS and the RMSE of the wind speed retrieved by the SST-dependent model were both improved. This indicates that the introduction of SST parameters can improve the accuracy of the model. This further suggests that while using Ka-band data to retrieve other sea surface parameters besides the winds, SST should also be considered; this will improve the accuracy of the corresponding model.

Our study used the DPR NRCS data and the collocated GMI wind speed and SST data. When retrieving wind speed, the collocated SST was applied to improve the accuracy of the model. Here, both DPR and GMI are carried on GPM and perfect collocated observation data can help reduce retrieval error. In this case, the wind speed error, driven by the incomplete spatio-temporal collocation, will not emerge. With the development of satellite remote sensing technology, an increasing number of satellites will carry various sensors for observation missions. The synergy of multiple sensor data on the same satellite to jointly retrieve sea surface parameters will be an important challenge in the future.

Despite the promising results we reported in this study, two considerable limitations should be mentioned. First, our study assumed that there was no impact of wind direction on NRCS. The combination of wind speed and wind direction can provide complete wind-field data. Some scholars have previously found that NRCS is sensitive to the wind

direction when the incidence angles is 8°–10° [25]; however, when the incidence angle is lower, the wind direction information was not pronounced and was often neglected due to this. Thus, the retrieval of the wind direction was not considered in this study. In addition, when studying the correlation between NRCS and CSPD, whether or not the different relative direction angles between the wind direction and the current direction could cause the change of the correlation coefficient is also worthy of follow-up research.

Second, the proposed model is not suitable for the data affected by rainfall. The inner beam and the outer beam correspond to different changes of NRCS caused by rainfall. For the scatterometer working at a medium incidence angle, NRCS will gradually increase when the rainfall rate gradually increases [26–29]. However, when the rainfall occurs, the radar NRCS at low incidence angles will generally become small [30–32]. Therefore, in this study, the data affected by rainfall were eliminated according to the rainfall identification during data preprocessing. The relationship between rainfall rate and NRCS based on TRMM PR Ku-band data has been examined in previous studies to correct the impact of rainfall on the data [33]; this means that the rain-corrected data are still promising for wind speed retrieval. In this case, one can correct the impact of rainfall on DPR Ka-band data to retrieve more abundant wind speed products.

## 5. Conclusions

In this study, a Ka-band wind speed retrieval model at low incidence angles was proposed. The data used consisted of GPM DPR, GMI wind speed, SST product, ECMWF SWH, and HYCOM CSPD products. First, the NRCS dependence on wind speed incidence angle, SST, SWH, and CSPD was analyzed. The NRCS, incidence angle, SST, and wind speed were used as the input in the analysis to establish the Ka-band wind speed retrieval model at a low incidence angle. The model was segmented for SST and second-order linear fitting was used to retrieve the model parameters. The SST-independent model was also established to understand the performance of the SST-dependent model. Then, the portion of data, independent of the fitting data, was applied to retrieve the sea surface wind speed, while it was ultimately validated by the GMI wind product. Finally, the BIAS and the RMSE of the SST-dependent and independent models at different incidence angles and SST were compared. Both models exhibited good wind speed retrieval accuracy, while the accuracy of the SST-dependent model was better with a 0.12 m/s improvement. Results showed that the proposed SST-dependent model can be qualified for wind speed retrieval from GPM DPR Ka-band data. We also believe that this model is suitable for measuring wind speed by other Ka-band spaceborne radars at low incidence angles.

**Author Contributions:** Conceptualization, C.J., L.R. and J.Y.; methodology, C.J., L.R. and Q.X.; validation, C.J. and J.D; writing—original draft preparation, C.J. and L.R.; writing—review and editing, C.J., L.R., J.Y., Q.X. and J.D. All authors have read and agreed to the published version of the manuscript.

**Funding:** This research was funded in part by the Zhejiang Provincial Natural Science Foundation of China (grant no.LGF21D060002), Project of State Key Laboratory of Satellite Ocean Environment Dynamics, Second Institute of Oceanography (grant no.SOEDZZ2205), the Scientific Research Fund of the Second Institute of Oceanography, Ministry of Natural Resources of China (grant no. JG1708), Project of Southern Marine Science and Engineering Guangdong Laboratory (Zhuhai) (grant no.311021004), and Natural Science Foundation of China (grand no.41976163).

**Institutional Review Board Statement:** Not applicable.

**Informed Consent Statement:** Not applicable.

**Data Availability Statement:** The DPR data used in this study are from NASA PPS data archive in https://arthurhou.pps.eosdis.nasa.gov, accessed on 1 November 2019. The GMI data were produced by Remote Sensing Systems and sponsored by NASA Earth Science funding. The data are available at www.remss.com, accessed on 10 December 2020. The global sea surface current speed data are from HYbrid Coordinate Ocean Model (https://www.hycom.org/, accessed on 16 December 2020)

and SWH data were from the European Center for Medium-Range Weather Forecasts (ECMWF) reanalysis datasets (https://www.ecmwf.int/, accessed on 16 December 2020).

**Acknowledgments:** The authors would like to thank the NASA Goddard Earth Sciences (GES) Data and Information Services Center (DISC) for providing the GPM DPR data, Remote Sensing Systems for providing the GPM GMI data, the European Center for Medium-Range Weather Forecasts (ECMWF) for the ERA-Interim data, and the HYbrid Coordinate Ocean Model (HYCOM) global sea surface current speed data. The authors would also like to thank four anonymous reviewers for their valuable comments to improve the manuscript.

**Conflicts of Interest:** The authors declare no conflict of interest.

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
