# Peer review of "Wind Speed Retrieval Using Global Precipitation Measurement Dual-Frequency Precipitation Radar Ka-Band Data at Low Incidence Angles"

_remotesensing, doi:10.3390/rs14061454_

Round 1
Reviewer 1 Report
General Comments
I recommend that this paper is accepted for publication with minor revisions
Presents a new SST and incidence angle dependent algorithm to retrieve wind speed data from precipitation radar Ka band data.
The development of the model is clearly explained, and appropriate data are used for validation. The results show good retrieval accuracy, which is better when SST dependency is taken into account.
Minor points
Line 20 Expand the acronym NRCS
Line 83 (and later): BIAS in capitals. Change to lower case?
Line 88: some missing words in this sentence?
“In view of the proposed models were successfully applied to precipitation radar data, other satellite data with the same incidence angle can also be applied.”
Maybe
“In view of the proposed models that have been successfully applied to precipitation radar data, other satellite data with the same incidence angle can also be applied.” ?
Line 103: Missing ° sign (±18°)
Line 172: Replace “Due to” with “As”. So the sentence reads “As both SWH and CSPD can cause changes of sea surface roughness, this study thus researched the NRCS dependence on SWH and CSPD.
Line 270: missing word? “Based (on) the developed model, this study used the lookup table method to retrieve the 270 wind speed
Reviewer 2 Report
The only advantage of these studies is that the remote methods used make it possible to obtain information from large areas in a fairly short time. But the measurement accuracy is small and significantly inferior to contact research methods. This article can be published, but after answering some questions.
- The accuracy of the working models must be compared with the results of contact measurements.
- At what altitude is the wind speed measured? It is known that near surfaces, the wind speed and direction are distorted due to turbulent processes.
- How does the size and mass of raindrops affect the measurement accuracy?
- How does the intensity of rain affect the accuracy of measurements?
Reviewer 3 Report
L 19-20: Sentence “The parameters that showed … “ is unnecessary – delete.
L 58: “infers” should be implies. This sentence is also somewhat unnecessary.
L 87-90: These 2 sentences need to be rephrased in better English to say something like: Assuming proper radiometric calibration, the previously derived models should apply to other radars with the same incidence angles.
L 99: Ref [15] should be referred to as Vandemark et al.
L 110: Replace “In this regard” with something like: In this paper we study …
L 127: This should be explicit that land data were not used. Was any border used around the land-flagged points?
L 128: How is the smoothing size related to the nominal footprint size and spacing? Was any overlap used? Was it a boxcar or some other smoothing?
L 139: Why were the model data not interpolated to the satellite data? That is what is usually done.
L 141: This assumes that there is no up/down wind (direction) effect. It also ignores the current direction. These should be stated, especially because of the comments above about those effects. There is a well-known scatterometer effect that the backscatter depends on the net wind speed relative to the water surface.
Fig 2: What is the WS bin size for the points on the plot? The symbols make it look like the bins were different for different incidence angles. Given that most models show an RMS error of ~1.5 m/s, it would seem that bins smaller than 0.5, or perhaps 0.25, m/s are not warranted.
L 166: “compatible” -> useful
L 172, 177: As noted above, the main effect of CSPD is to change the relative velocity of the wind and sea surface. The effect of currents directly on roughness would need justification, references.
Fig 3: What are the bin sizes on the axes? It seems that the plots should look pixelated, not so smooth. It would seem that at the largest incidence angle, the wave direction and/or the relative wind-wave direction might make a difference. A comment on this should be made.
The wind speed ranges seem too narrow.
My assessment of numbers per bin:
15e6 pt overall. 9 incid ~1.6e6/incid. Say 10 1 m/s WS ~160k/incid/WS. 0.5-4m SWH @ 0.5m ~ 20k to be divided among NRCS bins ~11-15 @0.2 dB?? ~ 1k/ bin. BUT legend says WS range was only 0.1m/s, so all numbers divided by 10 give ~100/bin.
(Also applies to L 210) Altimeters have shown that below 3 m/s NRCS has very wide scatter. Using 3 rather 4 m/s for the lowest WS seems likely to produce unreliable/uninteresting results.
None of these comments would change the conclusion that using SWH in the GMF would be difficult and not very helpful.
Fig 4: What are the bin sizes on the axes? Similar comments to above about number of points per bin. Also, see previous comments on CSPD.
Fig 5: Use same NRCS scale(s) on Figs 3-5.
L 231, Fig 6: How big are SST bins?
L 256-262: What fitting method is used?
Tables: What are the uncertainties (formal errors) on these coefficients? Without some idea of the uncertainty, the differences among the models cannot be assessed.
Fig 8: What are bin sizes? The “ears” on the distribution in (a) deserve some comment, as does the “kink”/relatively large deviation around 8 m/s. The correlation coefficient should be given.
Sec 4: The first paragraph would be a good replacement for the material starting at L87 in the Introduction. Suggest moving/combining it there.
L 356: What is the word “frequency” doing here?
L 387: There are numerous studies on rain and scatterometer data. A couple of additional citations should be given.
L 409: The link is incomplete/incorrect. How are these figures different than those included?
Round 2
Reviewer 1 Report
I have no additional comments to my first review. I am happy to recommend that this revised version is now suitable for publication
Author Response
Thank you for your review of this manuscript.
Reviewer 2 Report
In this form, the article can be accepted.
Author Response

(The authors gave the same response as above.)

Reviewer 3 Report
REVISED
Manuscript ID: remotesensing-1571326 Type of manuscript: Article
Title: Wind speed retrieval using Global Precipitation Measurement dual-frequency precipitation radar Ka-band data at low incidence angles
Authors: Chong Jiang, Lin Ren, Jingsong Yang *, Qing Xu, Jinyuan Dai
L 86: From previous paragraph, it seems that list of reference numbers should include 12, 13, so all together 11-16.
L 145: The fact that direction is not considered greatly reduces the relevance of any analysis of surface currents (L112). I do not see the material that the authors say they added in response #9. The point is not that wind direction in general is important at nadir (it is not or cannot be determined), but rather that if one is going to consider current speed then the direction of the wind relative to the current is important and the directions should not be ignored.
L 157: What is meaning of words “rail lifting”? What part of data were selected?
Fig 2,3: Since basically half the seasonal cycle was selected, there is a large expected change, especially above ~+/-45 deg in average wind speed winter/summer. Perhaps show distributions for WS, SST for January and June.
Fig 2: Depending on the time period, the low WS at 50-60S is somewhat unexpected.
Fig 4: This figure shows the well known fact from altimeters that at WS<~3m/s near-nadir NRCS is a poor or unreliable measure of WS. In some ways it is good that the authors selected WS down to 2 m/s to demonstrate that his holds at Ka-band, but other than a comment, the relationship in the 1.5-3 m/s range is not worth pursuing.
The analysis in Fig 5-7 would be much more meaningful done at WS of 4, 8, 12 (and 16 if there is enough data) +/-0.5 m/s.
L 203: I would disagree with the statement “when the wind speed is 3 m/s and the incidence angle is 1°, the NRCS decreases gradually with the increase of SWH. In other cases, the relationship between NRCS and SWH is not clear.” The densest part of the distribution goes the other way. The clearest correlation is what is discussed in the next comment (L 207).
L 207: The statement regarding incidence angle appears incorrect. The only really consistent behavior is for WS=7m/s where the correlation of ~-0.3 is . Except for WS=3m/s, the behavior at incidence=7deg is (basically) consistent with that at 1 and 4 deg.
Fig 5, 6, 7: The plots might give a somewhat different impression if they are all plotted on the same NRCS scale, say 6-18 dB. This comment was made previously and basically ignored in Response #15. I hope that even if not shown the authors have looked at this. Some text on what they found would be useful.
Response #13: I do not like this approach of smoothing the data and making the figures appear smoother. The data should be presented directly with bin sizes appropriate to the measurement accuracy and/or meaningful variation in the quantities of interest.
Fig 6: Is the WS range really +/-0.1 m/s? If so, the counts for what appear to be quite small bins in NRCS and SWH within the plot seem implausible, and the range for NRCS (~+/-0.5 m/s) for a small WS range would mean that WS errors from the measurement would likely be quite large. In that regard, SWH bins smaller than 0.2 m are not reasonable given likely data accuracy; the dots on the figures suggest that the bin size is much smaller. Either a comment that the dots are the centers of bins of x size or coloring the plot at the actual resolution is needed.
Fig 6: As discussed above regarding L 145, ignoring direction makes this analysis close to meaningless; or, at the very least, reiteration that direction has been ignored.
L 233: Should refer to Fig 7, not 5.
L234: This statement is related to above comments about nadir NRCS being a poor measurement of WS for WS<~3 m/s, so it is not very meaningful for discussing the relationship to SST.
L 240: Should refer to SST, not SWH.
L 255: Add as in Fig 8 caption “at and incidence angle of 4deg”.
L 277: Should refer to Fig 9, not 7.
L 283-4: “dependent” -> “dependence”
Fig 9: Without some indication of data density along the vertical lines this figure suggests that NRCS is not really a useful measurement of WS. For example, NRCS=10 intersects the vertical lines anywhere between about 7-14 m/s. Unless the figure is redone with distributions or there is some discussion of how few points there are near the ends of the lines, the figure should just be deleted as it appears to comprise the whole enterprise.
Fig 10, 11: Needs to note that same time period as Fig 2, and state explicitly again what that is. It would be much more useful to show one of these models, say the other is very similar and then show the difference between the model and the observations.
L 326, 335: Reference should be to Fig 12, not 8.
Fig 12: (b) appears to show data with GMI WS>16m/s while (a) does not ((a) cutoff may be even lower ~15.5). The GMI WS should not depend on which model is used, or if there is some lack of SST data (b) might cut off lower.
Response #18 and discussion in L 332-336: This addresses in only a very round about way the question of what is the uncertainty on the coefficients. If the formal uncertainties are such that the coefficients are indistinguishable, then different fitting intervals or fewer coefficients should be considered.
L 393, 395: Without again mentioning the lack of directional study, I do not think that CSPD can be dismissed.
Round 3
Reviewer 3 Report
The paper is greatly improved. The new binned plots are exactly what I requested in previous reviews.
Two small comments:
L 247: Should be 4 m/s rather than 3.
L 395: Title of Ref 19 suggests that it has a Ka-band model function. Is this statement really correct? Maybe previous did not have SST?